# Reliable Attribute-missing Multi-view Clustering with Instance-level and Feature-level Cooperative Imputation

## ABSTRACT

Multi-view clustering (MVC) constitutes a distinct approach to data mining within the field of machine learning. Due to limitations in the data collection process, missing attributes are frequently encountered. However, existing MVC methods primarily focus on missing instances, showing limited attention to missing attributes. A small number of studies employ the reconstruction of missing instances to address missing attributes, potentially overlooking the synergistic effects between the instance and feature spaces, which could lead to distorted imputation outcomes. Furthermore, current methods uniformly treat all missing attributes as zero values, thus failing to differentiate between real and technical zeroes, potentially resulting in data over-imputation. To mitigate these challenges, we introduce a novel Reliable Attribute-Missing Multi-View Clustering method (RAM-MVC). Specifically, feature reconstruction is utilized to address missing attributes, while similarity graphs are simultaneously constructed within the instance and feature spaces. By leveraging structural information from both spaces, RAM-MVC learns a high-quality feature reconstruction matrix during the joint optimization process. Additionally, we introduce a reliable imputation guidance module that distinguishes between real and technical attribute-missing events, enabling discriminative imputation. The proposed RAM-MVC method outperforms nine baseline methods, as evidenced by real-world experiments using single-cell multi-view data.

## CCS CONCEPTS

• **Computing methodologies** → **Cluster analysis**; • **Theory of computation** → **Unsupervised learning and clustering**;

## KEYWORDS

Multi-view Clustering, Multi-view Learning, Attribute-missing Imputation

## 1 INTRODUCTION

Multi-view clustering (MVC) represents a pivotal paradigm in machine learning, garnering widespread attention owing to its superior performance in uncovering data structures [7, 13, 33, 50, 54]. By aggregating information from multiple views, MVC offers a comprehensive understanding of the data and is extensively applied across a variety of real-world scenarios [10, 16, 22, 41, 45, 57], including

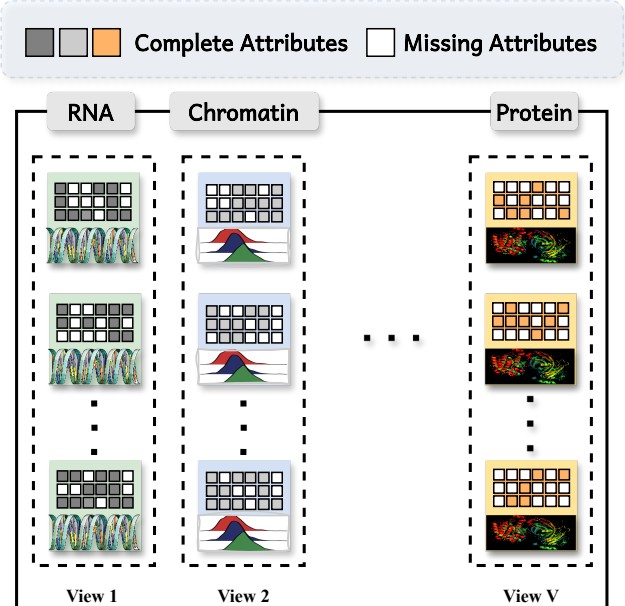

**Figure 1: Multi-view data exhibiting missing attributes, as exemplified in biomedical scenarios.**

biomedical research, social network analysis, and recommendation systems, etc. However, due to technical limitations of collection equipment, complex system environments, and privacy concerns, missing attributes in multi-view data remain a prevalent issue [1, 8]. For instance, within the public biological dataset [2], over 80% of the collected multi-modal sequencing results for various genes are missing, as is illustrated in Fig. 1. The missing attributes pose a significant challenge to existing MVC analysis.

Existing MVC methods predominantly address issues of instance-level missingness [18, 20, 21, 23], referring to incomplete instances within the observed view. Several studies introduce the concept of view-level missingness [12, 25, 39, 43], which is just a specific case of instance-level missingness that occurs when all samples in a view are missing. These studies typically develop a mapping function between complete and missing views to impute zero values in missing instances. For instance, Goeleven et al. developed separate mapping functions for each of the four complete views to impute information in target views with missing samples [11]. These mappings are constructed based on instance-level similarity, neglecting feature similarity and thus overlooking potential synergies between instance and feature spaces. Furthermore, in real-world multi-view data, cases of missing attributes occur far more frequently than those of missing instances. For example, in patients undergoing physical examinations, data on respiratory function, blood pressure, and electrocardiograms were collected,

while urine tests were inadvertently omitted. Existing MVC methods fail to account for missing attributes and are neither effective nor appropriate for addressing the attribute-missing issue. Merely applying the mapping function learned in the instance space for imputation can lead to distorted outcomes.

Furthermore, existing MVC methods address missing attributes indiscriminately [5, 38, 48, 52, 53, 56]. However, in practical applications, there are at least two scenarios for missing attributes: real and technical zero values. For instance, one student genuinely scored zero points on an exam, while another earned ninety-nine points but was assigned a score of zero due to the loss of his test paper. Non-discriminatory restoration of zero values can result in students who genuinely scored zero points receiving inflated scores, deviating from actual outcomes. This problem is especially prevalent in biomedical contexts. Single-cell multi-view data frequently feature a significant number of zero values [17, 26, 30]. Some of these zero values signify expression intensities with biological significance, whereas others stem from information loss attributable to technical detection limitations. Indiscriminate imputation can lead to unexpected and uninterpretable outcomes.

To address the aforementioned issues, we have developed a novel MVC framework for handling missing attributes, named Reliable Attribute-Missing Multi-View Clustering (RAM-MVC). This model comprises two main modules: 1) A reliable imputation guidance module, which assesses whether imputation should proceed based on the confidence levels of zero values and discriminates between real and technical zero values. 2) A bi-level cooperative imputation module that simultaneously extracts structural information from both feature and instance spaces, thereby enhancing the feature reconstruction process. Fig. 2 depicts the architecture of the proposed RAM-MVC framework. The contributions of this study are summarized as follows:

- We propose a pioneering unified attribute-missing MVC framework that seamlessly integrates bi-level imputation with reliable guidance, ensuring both components collaborate effectively to achieve accurate imputation and enhanced clustering performance.
- The developed imputation guidance module effectively differentiates between real and technical attribute-missing events, thus addressing the over-imputation problem prevalent in existing MVC methods. Additionally, by leveraging the structural information from both the instance and feature spaces, the Bi-level imputation module jointly optimizes the feature reconstruction matrix and secures high-quality features.
- An effective alternative optimization algorithm is designed to solve the proposed RAM-MVC model. Extensive experimental results demonstrate the model's superiority over other benchmark methods.

## 2 RELATED WORK

During the data collection process, a significant portion of multi-view data often experiences the loss of specific views or values due to technical limitations. The MVC method seeks to leverage the consistency and complementary of features across multiple views to address the missing data, thus mitigating the impact of incomplete data on clustering performance. In this section, we will categorize and review the existing missing MVC methods into two types: Instance-missing MVC Methods and Attribute-missing MVC Methods.

### 2.1 Instance-missing MVC Methods

In terms of instance absence, Liu et al. redefined the issue of missing instances as a challenge of completing incomplete view similarity graphs [28], successfully yielding discriminative representations. Zhang et al. introduced a two-step strategy that combines missing view imputation with hidden view learning to develop an interpretable model [55]. Wang et al. formulated a completion module leveraging cross-view relation transfer to infer missing data via graph networks [42]. Xu et al. introduced a mixed Gaussian prior and proposed a new strategy based on variational autoencoders, effectively aggregating information from multiple views and optimizing shared representation, thereby achieving improved clustering performance [46]. Additionally, Chao et al. proposed a novel incomplete contrastive multi-view clustering method recently. They utilized an attention mechanism to fuse samples and employed confidence levels to learn complementary information between each view. Furthermore, an end-to-end framework was designed to integrate multiple steps for joint optimization [6]. Liu et al. proposed an instance-level similarity graph learning method to enhance existing incomplete MVC methods [29]. They compressed all instances into a shared space, constructed cross-view similarities, and continuously optimized the constructed similarity matrix. This approach achieved excellent results in addressing the incomplete multi-view clustering challenge.

### 2.2 Attribute-missing MVC Methods

Regarding attribute absence, Peng et al. employed graph diffusion techniques to enhance the consistency of node embeddings across two views [35], thus facilitating attribute imputation within the input space of graphs. However, their method remains operational at the instance level, without the ability to modify attribute values. Pu et al. developed an adaptive imputation layer for handling missing multi-view data [36]; the layer constructs upon the results of soft clustering across multiple views, and is integrated with global cluster centroids. Nonetheless, it primarily addresses missing instances. Yu et al. introduced the concept of value-level missing. They performed a simple matrix correction to allow the existing MVC method to be applicable for incomplete MVC data, but did not delve into further research on missing attributes [51]. Different from these methods, Wu et al.'s team constructed a generative adversarial model based on optimal transmission theory to enhance the interpolation process [44]. This interpolation method was proven effective, yet further exploration of the clustering framework was not pursued. Despite these advancements, no multi-view clustering model has been specifically designed to address missing attributes. Furthermore, current methods neglect the potential synergistic effects between the instance and feature spaces.

## 3 PROPOSED METHOD

First, we provide a brief introduction to the mathematical notations used in our manuscript. The multi-view data comprises $n$ instances across $v$ views and can be mathematically represented

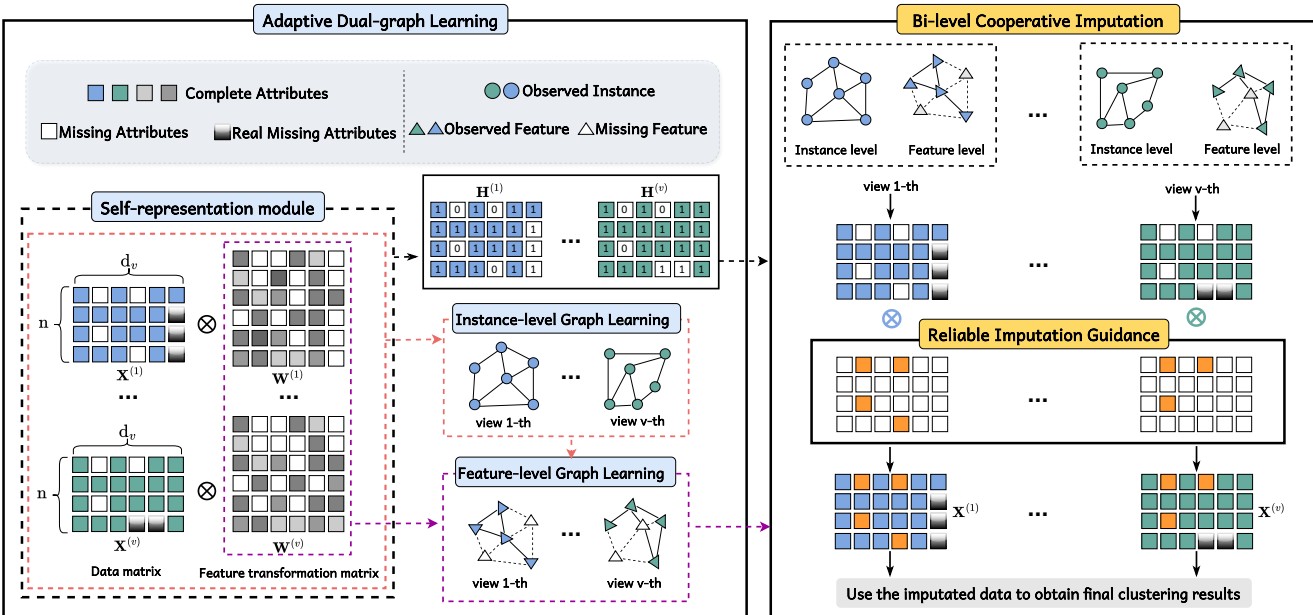

Figure 2: The framework for the proposed RAM-MVC. This model is composed of two main modules: 1) A Reliable Imputation Guidance Module that distinguishes between real and technical missing events during the imputation process. 2) A Bi-Level Cooperative Imputation Module that reconstructs features based on structural information from both the instance and feature spaces.

## Table 1: Symbol Appointment

| Symbol | Description |
|---|---|
| $d_p$ | The dimension in the $p$-th view. |
| $N$ | The number of the instances. |
| $z_j^{(k)}$ | The zero expression rate in the $k$-th cluster. |
| $m_j^{(k)}$ | The mean value in the $k$-th cluster. |
| $q_j^{2(k)}$ | The variance value in the $k$-th cluster. |
| $l_j^{(k)}$ | The confidence level of zero value. |
| $\mathbf{X}^{(p)} \in \mathbb{R}^{n \times d_p}$ | The raw data of the $p$-th view. |
| $\mathbf{H}^{(p)} \in \mathbb{R}^{n \times d_p}$ | The reliable imputation guidance matrix of the $p$-th view. |
| $\mathbf{W}^{(p)} \in \mathbb{R}^{d_p \times d_p}$ | The feature transforming matrix of the $p$-th view. |
| $\mathbf{S} \in \mathbb{R}^{n \times n}$ | The similarity matrix. |
| $\mathbf{L}_s \in \mathbb{R}^{n \times n}$ | The Laplacian matrix. |
| $.^{\top}$ | The transpose of matrix. |
| $\| \cdot \|_F$ | Frobenius norm. |
| $\mathrm{Tr}(\cdot)$ | The trace of matrix. |
| $\alpha_p$ | The weight coefficient of the $p$-th view. |
| $\lambda$ | Regularization parameters. |
| $t$ | Confidence threshold. |

as $\{\mathbf{X}^{(p)} \in \mathbb{R}^{n \times d_p}\}_{p=1}^{v}$, where $\mathbf{X}^{(p)} = \{\mathbf{x}_1^{(p)}; ...; \mathbf{x}_n^{(p)}\}$, and $d_p$ denotes the dimension in the $p$-th view. The Frobenius norm of $\mathbf{X}^{(p)}$ is expressed as $\|\mathbf{X}^{(p)}\|_F = \sqrt{\sum_{i=1}^{n} \sum_{j=1}^{d_p} (x_{ij}^{(p)})^2}$, while $\mathrm{Tr}(\mathbf{X}^{(p)})$ and $\mathbf{X}^{(p)\top}$ respectively represent the trace and transpose of $\mathbf{X}^{(p)}$. Symbols used in our paper can be referenced in Table 1.

## 3.1 Reliable Imputation Guidance Module

There are two potential types of attribute-missing events: 1) Real missing events, which refer to the non-expression of features in instances, resulting in zero values, and 2) Technical missing events, which refer to the loss of attributes during the data collection process due to limitations of technologies. Here, we aim to retain the real missing events as they provide insights into real-world facts, while recovering the technical zero values to address the information loss.

To differentiate between these two types of attribute missingness, confidence levels for each feature $l_j^{(k)}$ are computed according to scImpute [19]. Specifically, zero values for features exhibiting high expression and low variability within instance clusters are considered technical zeros and require recovery. Conversely, features that consistently exhibit low levels of expression and demonstrate high variability are identified as real zeros, reflecting important insight and requiring preservation. The formula for calculating the confidence levels $l_j^{(k)}$ is provided below:

$$l_j^{(k)} = \frac{\left(1 - z_j^{(k)}\right) m_j^{(k)}}{\left(1 - z_j^{(k)}\right) m_j^{(k)} + z_j^{(k)} q_j^{2(k)}}, \quad (1)$$

where $k$ represents the cluster number, while $z_j^{(k)}$, $m_j^{(k)}$, and $q_j^{2(k)}$ denote the zero expression rate, the mean value, and the variance value, respectively.

The reliable imputation guide matrix $\mathbf{H}$ is computed based on the confidence levels $l_j^{(k)}$. The process is as follows:

$$h_{ij} = \begin{cases} 1 & \text{if } x_{ij} > 0, \\ 1 & \text{if } l_j^{(k)} < t \in (0, 1), \\ 0 & \text{otherwise.} \end{cases} \quad (2)$$

where $t$ denotes the confidence threshold established to prevent excessive imputation. Guiding matrix $\mathbf{H}$ corresponds in dimensions to the numerical matrix $\mathbf{X}$, with elements populated exclusively by values of 0 and 1. These indicate whether missing values should be imputed at corresponding locations in $\mathbf{X}$. If the confidence level falls below $t$, a value in matrix $\mathbf{H}$ is set to 1, thus protecting the real zero values from imputing.

## 3.2 Feature-level Graph Learning

When missing features appear in the observed view, utilizing existing features to infer the missing ones represents a simple and intuitive idea. Therefore, we propose extending the concept of self-representation to the feature level. Specifically, we posit that each feature can be reconstructed via a linear combination of other features. Given the multi-view dataset $\mathbf{X}^{(p)} \in \mathbb{R}^{n \times d_p}$, we present the mathematical formulation as follows:

$$\min_{\mathbf{W}^{(p)\top}\mathbf{W}^{(p)}=\mathbf{I}} \sum_{p=1}^{v} \|\mathbf{X}^{(p)} - \mathbf{X}^{(p)}\mathbf{W}^{(p)}\|_F^2, \quad (3)$$

where $\mathbf{W}^{(p)} \in \mathbb{R}^{d_p \times d_p}$ denotes the projection matrix for the $p$-th view, used to map the original data to a new feature space. Each row in $\mathbf{W}^{(p)}$ signifies the contribution of each feature to the reconstruction, relative to other features. In essence, this matrix can illustrate the significance of each feature. Furthermore, orthogonal constraints $\mathbf{W}^{(p)\top}\mathbf{W}^{(p)} = \mathbf{I}$ are applied to $\mathbf{W}^{(p)}$ to promote feature independence, helping to eliminate redundant information among features. Moreover, limiting the complexity of the transformation matrix will constrain the model's capacity, thereby mitigating the risk of overfitting. In this study, we employ the matrix $\mathbf{W}^{(p)}$ to facilitate the reconstruction of missing features, thus, the quality of the reconstructed features is deeply contingent upon the quality of $\mathbf{W}^{(p)}$.

## 3.3 Instance-level Graph Learning

Numerous MVC methods focus on feature recovery by reconstructing instance-level information, with their limitations outlined in our prior discussion. However, these limitations should not be interpreted as a disregard for the importance of the structural information in the instance space. We maintain that structural information in both instance and feature spaces is of equal importance. Their joint optimization facilitates the capture of potential synergistic effects across domains, enabling the model to uncover deeper insights into the missing data. Therefore, in this section, we introduce the construction of structural information at the instance level.

Leveraging the foundational principles of spectral theory and manifold learning [9], the similarity between two instances in the original high-dimensional space signifies their proximity within a specific local neighborhood in that space. Consequently, this similarity ought to be preserved during the dimensional reduction of these data to a lower-dimensional space. In light of the aforementioned principles, we have devised a module to adaptively learn the nearest neighbor graph and obtain the structural information of the instance space, with the corresponding formula expressed as follows:

$$\min_{\mathbf{S}} \sum_{i,j=1}^{n} \|\mathbf{x}_{i,:}^{(p)}\mathbf{W}^{(p)} - \mathbf{x}_{j,:}^{(p)}\mathbf{W}^{(p)}\|_2^2 s_{ij}, \quad (4)$$
$$\text{s.t. } s_{ij} \geq 0, \mathbf{1}^\top \mathbf{s}_{:,i} = 1,$$

where $\mathbf{x}_{i,:}^{(p)}$ denotes the i-th row of $\mathbf{X}^{(p)}$, while $\mathbf{x}_{:,i}^{(p)}$ denotes the i-th column of $\mathbf{X}^{(p)}$. The constraints $s_{ij} \geq 0$ guarantee the rationality of the constructed similarity matrix, while the column constraints of $\mathbf{1}^\top \mathbf{s}_{:,i} = 1$ ensure that the elements of each column in the similarity matrix sum to 1, preventing single data points from disproportionately influencing the overall structure.

To facilitate a simpler solution, leveraging the properties of matrix trace, Eq. (4) can be transformed into an equivalent form for optimization:

$$\min_{\mathbf{S}} \text{Tr}\left(\mathbf{W}^{(p)\top}\mathbf{X}^{(p)\top}\mathbf{L_S}\mathbf{X}^{(p)}\mathbf{W}^{(p)}\right), \quad (5)$$
$$\text{s.t. } \mathbf{S} \geq 0, \mathbf{1}^\top \mathbf{S} = 1,$$

where $\mathbf{L_S} = \mathbf{I} - \mathbf{D}^{-\frac{1}{2}}\mathbf{S}\mathbf{D}^{-\frac{1}{2}}$ represents the Laplacian matrix, $\mathbf{S}$ represents the similarity matrix, and $\mathbf{D}$ represents the diagonal matrix derived from $\mathbf{S}$.

## 3.4 Bi-level Cooperative Imputation

Instance-level similarity has been extensively utilized in feature reconstruction and has achieved widespread verification [40, 49], whereas interpreting feature-level similarity presents challenges. To provide an intuitive understanding, we use single-cell multi-view data as an example, which regards genes as features and highlights a complex network of interactions among genes.

Therefore, we posit that structural information concurrently exists within both instance and feature spaces, and these two spaces possess the potential for synergistic effects. To adaptively learn the structural information from both the instance and feature spaces, we combine Eq. (3) and Eq. (5) to construct the bi-level objective function as follows:

$$\min_{\mathbf{W}^{(p)},\mathbf{S},\boldsymbol{\alpha}} \sum_{p=1}^{V} \|\mathbf{X}^{(p)} - \mathbf{X}^{(p)}\mathbf{W}^{(p)}\|_F^2 +$$
$$\sum_{p=1}^{V} \alpha_p^2 \text{Tr}\left(\mathbf{W}^{(p)\top}\mathbf{X}^{(p)\top}\mathbf{L_S}\mathbf{X}^{(p)}\mathbf{W}^{(p)}\right) + \lambda \|\mathbf{S}\|_F^2, \quad (6)$$
$$\text{s.t. } \mathbf{W}^{(p)\top}\mathbf{W}^{(p)} = \mathbf{I}, \mathbf{S} \geq 0, \mathbf{1}^\top \mathbf{S} = 1, \boldsymbol{\alpha} \geq 0, \boldsymbol{\alpha}^\top \mathbf{1} = 1,$$

where $\alpha_p$ denotes the weight coefficient of the $p$-th view, which is constructed to balance individual views. By optimizing Eq. (6), we can effectively learn the structural information at both the instance and feature levels, thus obtaining a more accurate $\mathbf{W}^{(p)}$. As we state in Section 3.2, the quality of the reconstructed features is deeply contingent upon the quality of $\mathbf{W}^{(p)}$. By enhancing the quality of $\mathbf{W}^{(p)}$, we ultimately obtain the high-quality reconstructed feature.

 

---

**Algorithm 1** Iterative Algorithm of RAM-MVC

---

**Input**: Attribute-miss multi-view data $\{\mathbf{X}^{(p)}\}_{p=1}^{v}$; the parameters $\alpha$ and $\lambda$; the threshold of imputation guidance $t$.

1: Initialize $\{\mathbf{W}^{(p)}\}_{p=1}^{v}$, $\boldsymbol{\alpha}$, and $\mathbf{S}$.
2: Calculate $\mathbf{H}^{(p)}$ via Eq. (2).
3: **while** not convergent **do**
4:     Update $\{\mathbf{W}^{(p)}\}_{p=1}^{v}$ via Algorithm 2;
5:     Update $\mathbf{S}$ by solving Eq. (13);
6:     Update $\boldsymbol{\alpha}$ by solving Eq. (16);
7: **end while**
8: Calculate $\{\hat{\mathbf{X}}^{(p)}\}_{p=1}^{v}$ via Eq. (7);
9: Calculate $\hat{\mathbf{S}}$ by solving Eq. (8);

**Output**: Perform spectral clustering on $\hat{\mathbf{S}}$ to obtain the final results.

---

**Algorithm 2** Algorithm of updating $\mathbf{W}^{(p)}$

---

1: Initialize $\mathbf{W}^{(p)*} = \mathbf{W}^{(p)}$.
2: **while** not convergent **do**
3:     $\mathbf{B}^{(p)} = 2\left(\gamma_{max}\mathbf{I} - \mathbf{A}^{(p)}\right)\mathbf{W}^{(p)*} + 2\mathbf{X}^{(p)\top}\mathbf{X}^{(p)}$;
4:     Perform SVD on $\mathbf{B}^{(p)}$ as $\mathbf{B}^{(p)} = \mathbf{U}^{\top}\Sigma\mathbf{V}$;
5:     $\mathbf{W}^{(p)*} = \mathbf{U}^{\top}\mathbf{V}$;
6: **end while**
7: **Output**: $\mathbf{W}^{(p)} = \mathbf{W}^{(p)*}$.

---

After obtaining high-quality reconstructed features, we refrain from directly imputing them into the corresponding missing features. As previously mentioned, this operation leads to over-imputation issues. Here, we employ the guiding matrix $\mathbf{H}$ to perform discriminatory imputation. The imputation process is computed as follows:

$$\hat{\mathbf{X}}^{(p)} = (1 - \mathbf{H}^{(p)}) \circ \mathbf{X}^{(p)}\mathbf{W}^{(p)} + \mathbf{X}^{(p)}, \quad (7)$$

where $\circ$ represents the Hadamard product, $\hat{\mathbf{X}}^{(p)}$ represents the data after imputation, embodying high-quality features and subsequently utilized for further clustering analysis. Specifically, we recompute the similarity as follows:

$$\min_{\hat{\mathbf{S}}} \sum_{p=1}^{V} \sum_{i,j=1}^{n} \|\hat{\mathbf{x}}_{i,:}^{(p)} - \hat{\mathbf{x}}_{j,:}^{(p)}\|_2^2 \hat{s}_{ij} + \lambda\|\hat{\mathbf{S}}\|_F^2, \quad (8)$$
$$\text{s.t. } \hat{\mathbf{S}} \geq 0, \mathbf{1}^{\top}\hat{\mathbf{S}} = \mathbf{1}.$$

The ultimate clustering result is generated by performing spectral clustering on the refined similarity matrix $\hat{\mathbf{S}}$, and the overall procedure is detailed in Algorithm (1).

### 3.5 Optimization

The optimization problem in Eq. (6) can be solved using alternating optimization methods, where in each iteration, one variable is fixed while the others are optimized, and this process continues iteratively until the objective function converges.

**Update $\mathbf{W}^{(p)}$ with $\mathbf{S}$ and $\boldsymbol{\alpha}$ fixed.** Fixing other variables, the subproblem concerning $\mathbf{W}^{(p)}$ can be rewritten as:

$$\min_{\mathbf{W}^{(p)}} \sum_{p=1}^{V} \|\mathbf{X}^{(p)} - \mathbf{X}^{(p)}\mathbf{W}^{(p)}\|_F^2 +$$
$$\sum_{p=1}^{V} \alpha_p^2 \text{Tr}\left(\mathbf{W}^{(p)\top}\mathbf{X}^{(p)\top}\mathbf{L_S}\mathbf{X}^{(p)}\mathbf{W}^{(p)}\right), \quad (9)$$
$$\text{s.t. } \mathbf{W}^{(p)\top}\mathbf{W}^{(p)} = \mathbf{I}.$$

By removing irrelevant terms, we independently optimize $\mathbf{W}^{(p)}$ on each view as follows:

$$\min_{\mathbf{W}^{(p)}} \text{Tr}\left(\mathbf{W}^{(p)\top}\mathbf{A}^{(p)}\mathbf{W}^{(p)} - 2\mathbf{X}^{(p)\top}\mathbf{X}^{(p)}\mathbf{W}^{(p)}\right), \quad (10)$$
$$\text{s.t. } \mathbf{W}^{(p)\top}\mathbf{W}^{(p)} = \mathbf{I},$$

where $\mathbf{A}^{(p)} = \left(\mathbf{X}^{(p)\top}\mathbf{X}^{(p)} + \alpha_p^2\mathbf{X}^{(p)\top}\mathbf{L_S}\mathbf{X}^{(p)}\right)$. The above equation can be relaxed as follows,

$$\max_{\mathbf{W}^{(p)}} \text{Tr}\left(\mathbf{W}^{(p)\top}\left(\gamma_{max}\mathbf{I} - \mathbf{A}^{(p)}\right)\mathbf{W}^{(p)} + 2\mathbf{X}^{(p)\top}\mathbf{X}^{(p)}\mathbf{W}^{(p)}\right),$$
$$\text{s.t. } \mathbf{W}^{(p)\top}\mathbf{W}^{(p)} = \mathbf{I}, \quad (11)$$

where $\gamma_{max}$ is the largest eigenvalue of $\mathbf{A}^{(p)}$. According to reference [34], Eq. (11) can be solved by optimizing the following problem iteratively,

$$\max_{\mathbf{W}^{(p)}} \text{Tr}\left(\mathbf{W}^{(p)\top}\mathbf{B}^{(p)}\right), \quad (12)$$
$$\text{s.t. } \mathbf{W}^{(p)\top}\mathbf{W}^{(p)} = \mathbf{I},$$

where $\mathbf{B}^{(p)} = 2\left(\gamma_{max}\mathbf{I} - \mathbf{A}^{(p)}\right)\mathbf{W}^{(p)*} + 2\mathbf{X}^{(p)\top}\mathbf{X}^{(p)}$ and $\mathbf{W}^{(p)*}$ denotes the optimal $\mathbf{W}^{(p)}$ in the last iteration. By performing SVD decomposition $\mathbf{B}^{(p)} = \mathbf{U}^{\top}\Sigma\mathbf{V}$, the optimal $\mathbf{W}^{(p)}$ at each iteration is $\mathbf{U}^{\top}\mathbf{V}$.

**Update $\mathbf{S}$ with $\mathbf{W}^{(p)}$ and $\boldsymbol{\alpha}$ fixed.** Fixing other variables, the subproblem concerning $\mathbf{S}$ can be rewritten as:

$$\min_{\mathbf{S}} \sum_{p=1}^{V} \alpha_p^2 \text{Tr}\left(\mathbf{W}^{(p)\top}\mathbf{X}^{(p)\top}\mathbf{L_S}\mathbf{X}^{(p)}\mathbf{W}^{(p)}\right) + \lambda\|\mathbf{S}\|_F^2, \quad (13)$$
$$\text{s.t. } \mathbf{S} \geq 0, \mathbf{1}^{\top}\mathbf{S} = \mathbf{1}.$$

The above function can be rewritten as follows,

$$\min_{\mathbf{s}_{ij}} \sum_{p=1}^{V} \sum_{i,j=1}^{n} \frac{1}{2}\alpha_p^2\|\mathbf{x}_{i,:}^{(p)}\mathbf{W}^{(p)} - \mathbf{x}_{j,:}^{(p)}\mathbf{W}^{(p)}\|_2^2 s_{ij} + \lambda s_{ij}^2, \quad (14)$$
$$\text{s.t. } s_{ij} \geq 0, \sum_{j=1}^{n} s_{ij} = 1.$$

Considering that each column of $\mathbf{S}$ is uncorrelated with each other. Denoting $\mathbf{s}_{:,j}$ as a vector with $s_{ij}$ to be the $j$-th element, Eq. (14) can be optimized in column form as follows,

$$\min_{\mathbf{s}_{:,j}} \frac{1}{2}\lambda\mathbf{s}_{:,j}^{\top}\mathbf{s}_{:,j} + \mathbf{e}_{:,j}^{\top}\mathbf{s}_{:,j}, s.t. \mathbf{s}_i \geq 0, \mathbf{s}_i^{\top}\mathbf{1} = 1, \quad (15)$$

where $\mathbf{e}_{i,j} = \frac{1}{4\lambda}\sum_{p=1}^{v}\alpha_p^2\|\mathbf{x}_{i,:}^{(p)}\mathbf{W}^{(p)} - \mathbf{x}_{j,:}^{(p)}\mathbf{W}^{(p)}\|_2^2$. Following [34], we can easily get the closed form solution of each $\mathbf{s}_i^{(p)}$.

**Update $\boldsymbol{\alpha}$ with $\mathbf{W}^{(p)}$ and S fixed.** Fixing other variables, the objective function with respect to $\boldsymbol{\alpha}$ can be formulated as

$$\min_{\boldsymbol{\alpha}} \sum_{p=1}^{v} \alpha_p^2 \mathbf{g}_p, s.t. \boldsymbol{\alpha}^\top \mathbf{1} = 1, \boldsymbol{\alpha} \geq 0, \qquad (16)$$

where $\mathbf{g}_p = \mathrm{Tr}\left(\mathbf{W}^{(p)^\top} \mathbf{X}^{(p)^\top} \mathbf{L_S} \mathbf{X}^{(p)} \mathbf{W}^{(p)}\right)$. We can obtain the optimal $\alpha_p$ by Cauchy-Buniakowsky-Schwarz inequality as

$$\alpha_p = \frac{\frac{1}{\mathbf{g}_p}}{\sum_{p=1}^{v} \frac{1}{\mathbf{g}_p}}. \qquad (17)$$

### 3.6 Discussion

**Time complexity analysis.** The time complexity of our proposed RAM-MVC is $O(\sum_{p=1}^{v} n^2 d_v + \sum_{p=1}^{v} n d_v^2 \tau + \sum_{p=1}^{v} d_v^3 \tau)$, where $\tau$ represents the number of iterations in Algorithm 2. Specifically, updating $\mathbf{W}^{(p)}$ costs $O(n^2 d_v + n d_v^2 \tau + d_v^3 \tau)$ totally. Calculating $\mathbf{A}^{(p)}$ needs $O(n^2 d_v)$ and performing matrix multiplication in each iteration of Algorithm 2 needs $O(n d_v^2 + d_v^3)$. The computational complexity of optimizing S is $O(\sum_{p=1}^{v} n^2 d_v)$ for matrix multiplication and $O(n^2)$ for solving Eq. (15). Updating $\boldsymbol{\alpha}$ costs $O(\sum_{p=1}^{v} n^2 d_v)$ for calculating $\mathbf{g}_p$.

**Space complexity analysis.** The main matrix variables required to be stored during the computation process of RAM-MVC include the raw data matrix $\{\mathbf{X}^{(p)}\}_{p=1}^{v}$, the imputation guidance matrix $\{\mathbf{H}^{(p)}\}_{p=1}^{v}$, the feature transformation matrix $\{\mathbf{W}^{(p)}\}_{p=1}^{v}$, the similarity matrix S, and the Laplacian matrix $\mathbf{L_S}$. Without considering the space occupied by vectors, the space complexity of RAM-MVC is $O(n^2 + \sum_{p=1}^{v}(n d_v + d_v^2))$.

## 4 EXPERIMENTS

### 4.1 Experimental Settings

**Benchmark Datasets.** Through investigation [19, 27], we find that single-cell multi-view data naturally encompass both technical and real missing events. Therefore, in this study, we have chosen to evaluate our model in a biomedical scenario. Six real-world single-cell multi-view datasets are involved: BMNC-I, BMNC-II, PBMC, SLN111, SMAGE-I, and SMAGE-II. Detailed information about these datasets is provided in Table 2. The BMNC dataset was sourced from the GEO database, which is a comprehensive archive of high-throughput gene expression data and genomic information [3]. The PBMC, SMAGE-3K, and SMAGE-10K datasets were obtained from the renowned biological database at www.10xgenomics.com. SLN111 originates from the work of Yosef et al. [31].

**Competitive Algorithms.** We compare the proposed RAM-MVC with the following state-of-the-art approaches, i.e., FMCNOF (Fast Multi-View Clustering via Nonnegative and Orthogonal Factorization) [47]; FastMICE (Fast Multi-View Clustering Via Ensembles: Towards Scalability, Superiority, and Simplicity) [14]; RMKM (Multi-View K-Means Clustering on Big Data) [4]; UOMVSC (Unified One-Step Multi-View Spectral Clustering) [37]; AMGL (Parameter-Free Auto-Weighted Multiple Graph Learning: A Framework for Multi-view Clustering and Semi-Supervised Classification) [32]; MSGL (Structured Graph Learning for Scalable Subspace Clustering: From Single View to Multiview.) [15]; DCCA (Deep cross-omics cycle

**Table 2: Attribute-missing multi-view datasets in our experiments**

| Dataset | Size | Views | Clusters | Dimensions |
|---------|------|-------|----------|------------|
| BMNC-I | 1728 | 2 | 5 | 1000/25 |
| BMNC-II | 1963 | 2 | 4 | 1000/25 |
| PBMC | 3762 | 2 | 16 | 1000/49 |
| SLN111 | 6018 | 2 | 10 | 1000/112 |
| SMAGE-I | 2585 | 2 | 14 | 2000/2000 |
| SMAGE-II | 11020 | 2 | 12 | 2000/2000 |

attention model for joint analysis of single-cell multi-omics data) [59]; scMDC ( Clustering of single-cell multi-omics data with a multimodal deep learning method) [24]; scMVAE (Deep-joint-learning analysis model of single cell transcriptome and open chromatin accessibility data) [58].

**Evaluation Metrics.** In this study, we focus on the accuracy of clustering results. Consequently, four widely used external clustering evaluation metrics are employed: Accuracy (ACC), Normalized Mutual Information (NMI), Purity, and Adjusted Rand Index (ARI).

**Training Settings.** For all competitive algorithms, searches were conducted within the recommended parameter space to select the parameter combinations for optimal clustering performance. In our study, the $\lambda$ parameter values were set to $[1, 10, 100, 1000]$, while the threshold $t$ values were set to $[0.1, 0.3, 0.5, 0.7, 0.9]$. The experiments were repeated multiple times to accurately determine both the mean and the standard deviation. All trials were conducted on a Linux workstation equipped with an Intel Core i9-12900KF CPU and 64GB of RAM.

### 4.2 Performance Comparison

Table 3 presents the clustering performance of the RAM-MVC algorithm alongside nine baseline methods across six benchmark datasets. It is evident that RAM-MVC consistently achieves the optimal scores in most cases. In a total of 24 comparisons, RAM-MVC won first place in 58.33% of cases and was placed in the top two in 91.67% of instances. For datasets comprising over 10,000 instances, such as SMAGE-II, the RAM-MVC algorithm demonstrated superior performance relative to the average score, with an increase of (26.1%, 12.68%, 12.52%, 29.04%) in the ACC, NMI, Purity, and ARI metrics, respectively. For small datasets, such as BMNC-I and BMNC-II, in comparison with the average score, the RAM-MVC algorithm maintained its superior performance over comparative algorithms, with improvements in the four metrics (29.7%, 24.73%, 17.45%, 39.93%) and (23.5%, 26.39%, 16.66%, 31.5%), respectively. This observation highlights the RAM-MVC algorithm's consistent excellence across datasets of varying sizes. Moreover, it was found that underperformance occurs exclusively in the Purity metric. In analyzing the potential causes, the hypothesis is that class imbalance might be the underlying issue. For instance, assigning all instances to a single cluster results in a Purity value of 1; yet, this outcome does not equate to effective clustering. In terms of running time, our method does not offer a significant advantage. The comparison of running times of different models is detailed in Table 4.

**Table 3: Clustering performance on six benchmark datasets, the top 2 scores highlighted in bold.**

| Methods | FMCNOF | FastMICE | RMKM | UOMVSC | AMGL | MSGL | DCCA | scMDC | scMVAE | RAM-MVC |
|---|---|---|---|---|---|---|---|---|---|---|
| | | | | | ACC(%) | | | | | |
| BMNC-I | 53.41±0.00 | **78.62±2.25** | 61.81±0.00 | 72.97±0.00 | 21.68±0.09 | 70.62±10.99 | 71.30±0.00 | 68.55±2.11 | 65.42±0.09 | **92.41±0.03** |
| BMNC-II | 66.02±0.00 | 94.95±5.48 | 61.90±0.00 | 98.42±0.00 | 26.08±0.13 | 91.61±7.14 | **98.93± 0.00** | 79.54±1.14 | 63.73±0.00 | **99.19±0.00** |
| PBMC | 35.83±0.00 | 66.23±1.74 | **71.32±0.00** | 70.79±0.00 | 8.50±0.11 | 65.39±2.85 | 64.49±0.00 | 70.85±3.39 | 61.35±0.00 | **71.16 ±0.51** |
| SLN111 | 56.23±0.00 | 49.23±4.06 | 53.77±0.00 | **81.11±0.00** | 11.65±0.29 | 51.73±5.51 | 52.43±0.00 | 64.70±6.25 | 52.88±2.70 | **84.96±0.66** |
| SMAGE-I | 43.98±0.00 | 47.93±1.70 | 62.44±0.00 | 61.59±0.00 | 9.32±0.17 | 55.53±2.57 | 48.78±0.00 | **65.62±1.88** | 45.69±0.00 | **70.75±0.15** |
| SMAGE-II | 67.35±0.00 | 50.53±3.14 | 57.85±0.00 | **80.26±0.00** | 9.36±0.10 | 53.08±2.24 | 47.21±0.00 | 58.32±1.35 | 42.98±0.00 | **77.98±0.03** |
| | | | | | NMI(%) | | | | | |
| BMNC-I | 50.29±0.00 | **75.60±1.38** | 56.33±0.00 | 73.61±0.00 | 32.00±0.03 | 69.05±6.67 | 60.35±0.00 | 74.70±0.29 | 68.19±0.53 | **86.97±0.05** |
| BMNC-II | 55.47±0.00 | 75.93±1.16 | 70.79±0.00 | 93.11±0.00 | 0.30±0.01 | 84.90±5.82 | **94.74±0.00** | 74.80±1.79 | 71.83±0.00 | **95.49±0.00** |
| PBMC | 44.04±0.00 | 70.19±1.15 | 69.63±0.00 | **73.05±0.00** | 1.30±0.04 | 66.58±1.41 | 70.25±0.00 | 72.30±1.36 | 68.50±0.00 | **72.51 ±0.24** |
| SLN111 | 41.98±0.00 | 65.25±2.43 | 67.89±0.00 | **79.33±0.00** | 0.66±0.13 | 60.27±3.18 | 66.49±0.00 | 71.74±3.98 | 66.78±0.07 | **81.14±0.60** |
| SMAGE-I | 38.08±0.00 | 56.01±1.02 | 60.92±0.00 | 60.74±0.00 | 1.83±0.12 | 52.11±1.86 | 53.34±0.00 | **61.76±0.81** | 53.94±0.00 | **62.05±0.16** |
| SMAGE-II | 46.97±0.00 | 57.25±1.17 | 60.16±0.00 | **68.73±0.00** | 0.34±0.02 | 54.68±1.08 | 54.63±0.00 | 59.80±0.52 | 54.80±0.00 | **63.50±0.08** |
| | | | | | Purity(%) | | | | | |
| BMNC−I | 71.53±0.00 | **91.13±0.61** | 81.31±0.00 | 90.97±0.00 | 21.81±0.14 | 86.65±5.15 | 71.30±0.00 | 87.30±2.00 | 72.66±0.09 | **92.41±0.03** |
| BMNC-II | 77.99±0.00 | 91.26±5.83 | 92.56±0.00 | 98.42±0.00 | 26.18±0.14 | 95.31±2.37 | **98.93±0.00** | 92.10±0.15 | 70.05±0.00 | **99.19±0.00** |
| PBMC | 41.23±0.00 | 80.19±1.73 | 77.17±0.00 | **81.63±0.00** | 8.73±0.12 | 74.98±1.76 | 65.39±0.00 | 78.30±2.81 | 62.17±0.00 | **80.80 ±0.18** |
| SLN111 | 56.23±0.00 | 85.39±1.31 | 85.66±0.00 | **88.32±0.00** | 11.77±0.32 | 79.04±1.74 | 56.46±0.00 | 84.50±5.15 | 56.09±3.47 | **86.28±0.80** |
| SMAGE-I | 65.49±0.00 | 77.94±0.67 | **78.69±0.00** | 78.34±0.00 | 9.59±0.16 | 71.11±1.73 | 50.83±0.00 | **78.93±0.37** | 48.63±0.00 | 72.69±0.07 |
| SMAGE-II | 72.59±0.00 | 82.28±0.97 | **85.39±0.00** | 83.64±0.00 | 9.41±0.10 | 78.92±0.30 | 51.51±0.00 | 82.86±0.22 | 46.58±0.00 | 78.43±0.03 |
| | | | | | ARI(%) | | | | | |
| BMNC-I | 31.62±0.00 | **58.77±1.47** | 44.11±0.00 | 55.39±0.00 | 0.04±0.03 | 49.89±10.95 | 49.34±0.00 | 55.19±1.51 | 45.62±0.18 | **83.26±0.03** |
| BMNC-II | 55.45±0.00 | 93.67±6.54 | 52.02±0.00 | 97.36±0.00 | 0.06±0.01 | 86.61±4.37 | **97.93±0.00** | 62.44±4.18 | 52.62±0.00 | **97.96±0.00** |
| PBMC | 19.09±0.00 | 56.13±2.13 | 59.10±0.00 | **60.22±0.00** | 0.01±0.02 | 53.12±2.05 | 55.10±0.00 | 59.79±4.56 | 51.95±0.00 | **60.90 ±0.56** |
| SLN111 | 28.42±0.00 | 41.50±4.52 | 43.82±0.00 | **81.98±0.00** | 0.05±0.05 | 40.99±7.48 | 45.50±0.00 | 55.79±9.04 | 45.61±2.09 | **84.40±0.76** |
| SMAGE-I | 29.56±0.00 | 32.44±1.65 | 47.11±0.00 | 49.57±0.00 | 0.02±0.02 | 38.77±3.28 | 38.17±0.00 | **52.02±1.28** | 30.08±0.00 | **57.52±0.38** |
| SMAGE-II | 50.52±0.00 | 37.38±2.97 | 45.57±0.00 | **74.24±0.00** | 0.01±0.00 | 41.23±3.19 | 38.81±0.00 | 46.52±2.57 | 29.59±0.00 | **69.47±0.05** |

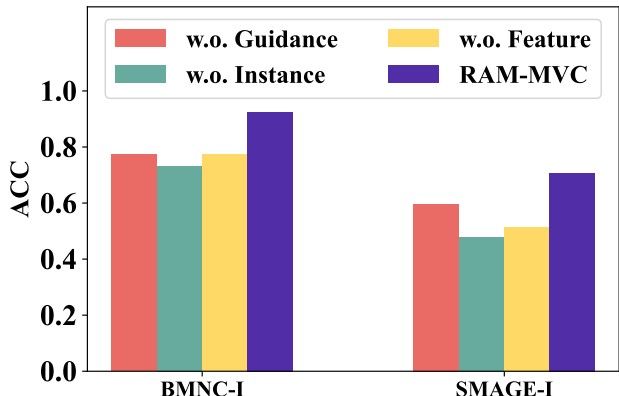

Figure 3: The ablation study was conducted on the proposed three variants: 1) w.o. Feature, 2) w.o. Instance, and 3) w.o. Guidance across the BMNC-I and SMAGE-I datasets, visualized using ACC metrics. (w.o. denotes without).

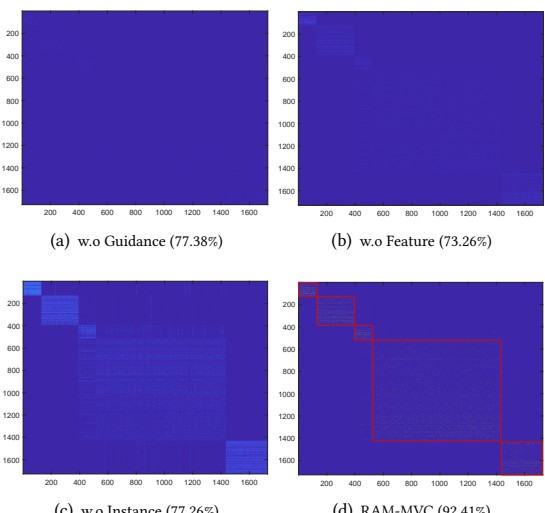

(a) w.o Guidance (77.38%)     (b) w.o Feature (73.26%)

(c) w.o Instance (77.26%)     (d) RAM-MVC (92.41%)

Figure 4: Illustrations of learned similarity matrix on BMNC-I datasets.

**Table 4: The comparison of running times (s) of different algorithms**

| Datasets | BMNC-I | BMNC-II | PBMC | SLN111 | SMAGE-I | SMAGE-II |
|---|---|---|---|---|---|---|
| FMCNOF | 0.20 | 0.19 | 0.40 | 0.48 | 0.66 | 3.03 |
| FastMICE | 2.62 | 2.77 | 3.90 | 5.70 | 7.32 | 15.73 |
| RMKM | 2.53 | 2.89 | 10.97 | 22.06 | 13.25 | 84.67 |
| UOMVSC | 0.92 | 1.09 | 8.79 | 27.02 | 4.37 | 129.58 |
| AMGL | 15.87 | 22.14 | 178.73 | 1207.77 | 27.80 | 4203.80 |
| MSGL | 3.23 | 4.97 | 9.56 | 38.45 | 21.03 | 59.18 |
| DCCA | 16.58 | 10.87 | 23.80 | 61.44 | 47.65 | 110.92 |
| scMDC | 108.15 | 170.60 | 101.46 | 424.02 | 40.67 | 149.78 |
| scMVAE | 296.19 | 262.61 | 28.81 | 363.81 | 72.82 | 178.54 |
| RAM-MVC | 13.45 | 15.05 | 98.67 | 441.00 | 52.47 | 3731.20 |

## 4.3 Ablation Study

To evaluate the effectiveness of the proposed module, we constructed three variants of RAM-MVC as follows: (1) RAM-MVC w.o. Feature: The feature-level graph learning module was removed from the complete model; (2) RAM-MVC w.o. Instance: The instance-level graph learning module was removed from the complete model; (3) RAM-MVC w.o. Guidance: The reliable imputation guidance module was removed from the complete model.

Fig. 3 presents the results of ablation experiments on BMNC-I and SMAGE-I datasets. We found that the variant RAM-MVC w.o. Guidance experienced significant performance degradation across two datasets, illustrating that the proposed reliable guidance module effectively addresses the discriminative imputation issue, thereby enhancing the clustering performance. Additionally, removing the instance-level and feature-level learning modules significantly degrades the model's performance to varying degrees, which illustrates the effectiveness of the bi-level cooperative imputation module. Fig. 4 presents the similarity matrix corresponding to various variants, indicating that preserving all modules achieves the highest-quality similarity matrix. To summarize, the ablation experiments underscore the effectiveness of our proposed three modules and highlight the superiority of the unified clustering framework integrating reliable guidance and bi-level cooperative imputation.

## 4.4 Parameter Sensitivity and Convergence Analysis

According to the object function in Eq. (6), the regularization hyperparameter $\lambda$ is incorporated, while the confidence threshold $t$ is introduced in Eq. (2). Thus, the RAM-MVC model incorporates two hyperparameters: $\lambda$ and $t$. To investigate the impact of hyperparameters on model performance, comprehensive parameter experiments were conducted on the BMNC-I and SMAGE-I datasets, the result is shown in Fig. 5. It is observed that across two datasets, fluctuations in $\lambda$ significantly affect clustering performance, yet stability can be maintained within a certain range. This indicates that RAM-MVC is sensitive to the $\lambda$ parameter, requiring adjustments within an appropriate range for optimal performance. Conversely, $t$ does not exhibit the same level of sensitivity as $\lambda$. On the SMAGE-I dataset, the parameter $t$ has a limited impact on the final performance. However, for the BMNC-I dataset, significant performance

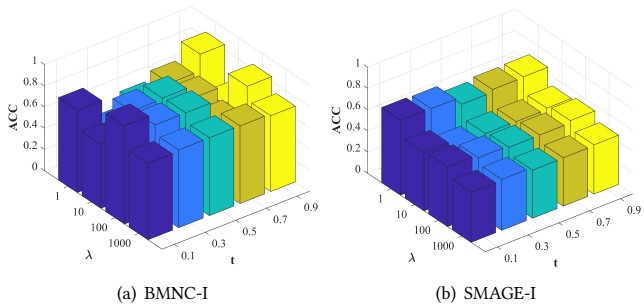

(a) BMNC-I  (b) SMAGE-I

**Figure 5: Parameter sensitive analysis of the proposed RAM-MVC on BMNC-I and SMAGE-I datasets.**

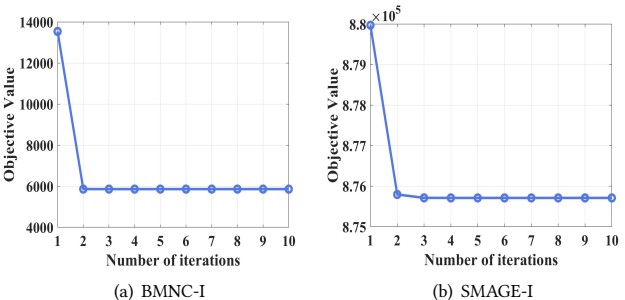

(a) BMNC-I  (b) SMAGE-I

**Figure 6: Objective function values at each iteration of the RAM-MVC on BMNC-I and SMAGE-I datasets.**

fluctuations were observed, demonstrating that the sensitivity of the parameter $t$ is related to the dataset's characteristics.

Furthermore, to investigate the convergence of RAM-MVC, convergence curves for the BMNC-I and SMAGE-I datasets were plotted, as depicted in Fig. 6. From the curves, it can be observed that the RAM-MVC algorithm converges after multiple iterations.

## 5 CONCLUSIONS

In summary, we propose a novel MVC method tailored for attribute-missing events, which overcomes the limitations of current methods that indiscriminately treat all missing attributes as zero values and neglect the contributions of bi-level structural information to feature reconstruction. The proposed RAM-MVC model seamlessly integrates reliable guidance and bi-level imputation into a unified learning framework. Instance-level and feature-level structural information is simultaneously leveraged to generate high-quality reconstructed features, and the confidence of zero values is calculated to facilitate discriminatory imputation on missing information, thereby avoiding over-interpolation. Furthermore, we have explored the application of this method in the biomedical field, demonstrating that it effectively completes the missing attribute information in single-cell multi-view data and achieves enhanced clustering performance. Experimental results from six real-world datasets underscore RAM-MVC's superiority over other benchmark methods.

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
