# OpenReview forum: "Reliable Attribute-missing Multi-view Clustering with Instance-level and feature-level Cooperative Imputation"
_acmmm.org/ACMMM/2024/Conference — MM2024 Poster_

### Official Review · Reviewer_v4QV · 2024-05-21

**Rating:** 3
**Confidence:** 4

**Summary:**

The paper introduces a novel algorithm, the Reliable Attribute-Missing Multi-View Clustering method (RAM-MVC), designed specifically to address missing attribute issues in multi-view data. The method focuses on distinguishing between true zeros and technical zeros during missing value imputation to avoid over-imputation. It combines a reliable imputation guidance module with dual-layer collaborative imputation to enhance feature reconstruction and clustering performance. Furthermore, experiments conducted on real-world single-cell multi-view datasets validate the effectiveness of the proposed method.

**Strengths:**

（1） the paper is well-structured；
（2）the methods and experimental results are easily understandable。

**Limitations:**

(1) While the authors provide a detailed explanation of "missing attributes" in the main text, it is advisable to briefly introduce this concept in the introduction for reader comprehension.

(2) Although the experiments evaluate the model in a biomedical scenario, demonstrating its effectiveness in this domain, further experimentation with datasets from other fields would elucidate RAM-MVC's performance across different domains and potential challenges, thus enhancing its generalizability.

(3) Figure 4, depicting similarity matrices corresponding to various variants in the ablation study, may not effectively convey the matrices for each experiment due to color choices. It is recommended that the authors use a different form of visualization, such as using contrasting colors with clear contrast, to enhance clarity.

(4) In the analysis of experimental results, the authors seem to lack specific analysis of the results related to runtime. Adding a section to analyze the reasons behind the observed runtime results would enhance the manuscript.

**Suitability:**

2

---

### Official Review · Reviewer_t6CX · 2024-05-22

**Rating:** 5
**Confidence:** 4

**Summary:**

This study introduces a novel approach to the Multi-View Clustering model, RAM-MVC. The method addresses a significant problem in existing MVC methods, namely, the omission of missing attributes. The authors focus on the synergistic effects between the instance and feature spaces to avoid distorted imputation outcomes. Additionally, the authors propose differentiating between real and technical zeros to prevent over-imputation. This work is insightful, and the experiments rigorously demonstrate the effectiveness of the algorithm.

**Strengths:**

1.The manuscript is excellently crafted, offering clear and direct explanations that facilitate easy understanding. The concepts presented are detailed, ensuring that they are communicated with precision and clarity.
2.The implementation of the reliable guidance module alongside the bi-level imputation module is notably impressive and highly effective. These components make a substantial contribution to the performance of the model.
3.The experimental section is extraordinarily thorough, encompassing detailed ablation studies that examine the contributions of different model components. Furthermore, the results not only meet but significantly exceed the current state-of-the-art benchmarks, demonstrating the model's superior performance and the effectiveness of the proposed methodology.

**Limitations:**

1.The method for addressing incomplete samples is well-established in existing MVC and is no longer considered a novel technology. During the process of sample imputation, features are also reconstructed. Consequently, I am curious about the advantages of the attribute imputation method proposed by the author.
2. There are some unclear aspects in the description of the experimental setup, such as the generation of the attribute-missing dataset and the definitions of technical and true missing events. Why is the algorithm evaluated in a single-cell scenario? What do technical and true missing represent in single-cell data, and what is their real-world significance?
3.Differentiating between real and technical attributes that are missing in multiview scenarios is interesting, and it would be beneficial if the authors could add and discuss relevant related work.
4.There remains room for enhancement in certain details to improve the quality of the paper. For instance, there appear to be some typographical errors in the "Time Complexity Analysis" section. I would suggest correcting it.
5. Equations (8) and (4) appear to have similar structures. Are these calculations performed twice?

**Suitability:**

3

---

### Official Review · Reviewer_J7Fj · 2024-05-23

**Rating:** 5
**Confidence:** 3

**Summary:**

The authors have developed an innovative Multi-View Clustering method that adeptly addresses the challenge of missing attributes. This represents a niche research direction, yet it is interesting. This model includes an adaptive dual-graph learning module specifically designed to preserve the underlying topological structure across varied feature spaces. Furthermore, the authors have developed an imputation guidance strategy that assesses the confidence of zero values, facilitating a precise approach to the imputation of missing data while effectively reducing the risk of over-interpolation. Empirical evidence from the analysis of six real-world datasets conclusively affirms the superior efficacy of the proposed RAM-MVC method.

**Strengths:**

- This paper significantly advances the field of attribute-missing multi-view clustering, offering insightful solutions that greatly enhance our understanding and capabilities in handling missing attributes.
- The paper introduces a novel concept that combines instance-level and feature-level graphs, clearly delineating the difference between technical and real zero values, thus enhancing the accuracy and interpretability of multi-view clustering methods.
- The paper features high-quality figures that are both informative and visually appealing. Additionally, the text is precise and clear, making the complex subject matter accessible.

**Limitations:**

-The dimensions of different datasets as presented in Table 2 appear to exhibit a degree of inconsistency, which prompts curiosity about the methodology the authors employed to ascertain the data dimensions across various views. The authors should publish a detailed approach to data preprocessing and feature selection.
-According to Fig. 5, the constructed temperature parameter t appears to have a relatively limited impact on the outcomes, suggesting that the model's sensitivity to this variable might be lower than anticipated, which warrants further investigation into its role in the algorithm's performance.
-The authors employ a biomedical multi-view dataset to assess the efficacy of their proposed model. However, the paper could be enhanced by providing a more thorough introduction to the dataset's background, detailing its composition, relevance, and the specific challenges it presents in the context of multi-view clustering.
-In Section 3.6, is it necessary to replace the symbol dv with dp? The author should check it.
- I am a little curious why the datasets involved in this article are almost two-views. Please explain it.

**Suitability:**

3

---

### Official Review · Reviewer_8B9a · 2024-05-25

**Rating:** 3
**Confidence:** 3

**Summary:**

The emphasis of this work is on complementing missing features through the cooperation on both the instance and feature levels among multiple views. This is an aspect that previous researchers did not touch upon, making it the innovative part of this work. Additionally, this article is rich in experiments and the theoretical proofs are also well-founded. However, there are still some issues with this paper.

**Strengths:**

1.This paper complements missing features through the cooperation on both the instance and feature levels among multiple views, which is an angle that previous works have not touched upon.
2.This work addresses the shortcomings of existing multi-view clustering methods in handling missing attribute data and provides an effective solution for the problem of missing attribute multi-view clustering.
3.A new optimization method has been utilized to address this problem.

**Limitations:**

1.At the feature level, the author used self-representation, which seems to be a common method. The method used in the Reliable Imputation Guidance Module part also directly applied an existing method, which does not reflect the author's sufficient innovation.
2.The number of views in the dataset is too small, just 2, which does not quite meet the related settings of multi-view, thus the persuasiveness of the experiment is somewhat lacking.
3.In the experimental part, the methods compared seem to have not compared the methods mentioned in the related work. The methods compared are all those not involving Instance-missing and Attribute-missing.
4.There is no detailed explanation as to why the Guiding matrix H is used. Can you provide specific literature references for this?
5.The RMKM and AMGL baseline methods are considered outdated.

**Suitability:**

3

---

### Meta-Review · Area_Chair_kfEn · 2024-07-04

**Recommendation:** Accept (Poster)
**Confidence:** 5

**Metareview:**

The paper received four reviews. After the rebuttal phase, the two reviewers who commented abut the rebuttal document stated that their doubts had been clarified,  and at the end one reviewer recommends borderline accept, one accept, one weak accept, and one borderline reject.
Given the fact that 3 reviewers out of 4 tend to be positive,  and the most negative one (still not so negative, borderline reject) didn’t assess the paper after the rebuttal, I recommend accepting the paper for poster presentation.